# Parvovirus B19 and Human Parvovirus 4 Encode Similar Proteins in a Reading Frame Overlapping the VP1 Capsid Gene

**DOI:** 10.3390/v16020191

**Published:** 2024-01-26

**Authors:** David G. Karlin

**Affiliations:** 1Division Phytomedicine, Thaer-Institute of Agricultural and Horticultural Sciences, Humboldt-Universität zu Berlin, Lentzeallee 55/57, D-14195 Berlin, Germany; davidgkarlin@gmail.com; 2Independent Researcher, 13000 Marseille, France

**Keywords:** overlapping genes, overlapping coding sequences, overlapping reading frames, 9 kDa protein, *erythroparvovirus*, *tetraparvovirus*, Y region, Y coding sequence, MAAP, non-coding functional element

## Abstract

Viruses frequently contain overlapping genes, which encode functionally unrelated proteins from the same DNA or RNA region but in different reading frames. Yet, overlapping genes are often overlooked during genome annotation, in particular in DNA viruses. Here we looked for the presence of overlapping genes likely to encode a functional protein in human parvovirus B19 (genus *Erythroparvovirus*), using an experimentally validated software, Synplot2. Synplot2 detected an open reading frame, X, conserved in all erythroparvoviruses, which overlaps the VP1 capsid gene and is under highly significant selection pressure. In a related virus, human parvovirus 4 (genus *Tetraparvovirus*), Synplot2 also detected an open reading frame under highly significant selection pressure, ARF1, which overlaps the VP1 gene and is conserved in all tetraparvoviruses. These findings provide compelling evidence that the X and ARF1 proteins must be expressed and functional. X and ARF1 have the exact same location (they overlap the region of the VP1 gene encoding the phospholipase A2 domain), are both in the same frame (+1) with respect to the VP1 frame, and encode proteins with similar predicted properties, including a central transmembrane region. Further studies will be needed to determine whether they have a common origin and similar function. X and ARF1 are probably translated either from a polycistronic mRNA by a non-canonical mechanism, or from an unmapped monocistronic mRNA. Finally, we also discovered proteins predicted to be expressed from a frame overlapping VP1 in other species related to parvovirus B19: porcine parvovirus 2 (Z protein) and bovine parvovirus 3 (X-like protein).

## 1. Introduction

Parvoviruses are small, non-enveloped viruses (for reviews, see [1,2,3]). Here we focus on two in particular: human parvovirus B19 (B19V) and human parvovirus 4 (PARV4). B19V causes several diseases in humans, such as fifth disease in children, cardiomyopathy, and persistent anemia in immunocompromised persons [4]. PARV4 is not formally associated with any disease, despite suspicions that it may cause encephalitis or accelerate HIV progression [5]. B19V and PARV4, respectively, belong to the genera *Erythroparvovirus* and *Tetraparvovirus* [2]; other species in these genera infect a variety of mammals.

The genome of every erythro- and tetraparvovirus encodes at least two proteins: the replicase NS1 and the capsid protein, of which at least two isoforms are made: VP1 and VP2 (Figure 1). In B19V, three additional ORFs (open reading frames) have been reported (Figure 1A): the 7.5 kDa ORF overlaps the NS1 ORF; the X ORF (which has the potential to code for a 9 kDa protein) overlaps the VP1 ORF; and the 11 kDa ORF partially overlaps the 3′ region of the VP1 ORF. The expression of the 7.5 kDa protein [6] and of the 11 kDa protein [7,8] have been proven experimentally. In contrast, the expression of the X protein has never been confirmed in infected cells. A substitution meant to knock out the expression of the X ORF caused no discernable change in viral replication or infectivity [9], raising doubts on the expression or functionality of the X protein.

Likewise, in PARV4, two ORFs overlapping the VP1 ORF have been noticed, but never confirmed experimentally [10]: ARF1 and ARF2 (ARF stands for “Alternative Reading Frame) (Figure 1B).

Overlapping ORFs are frequently overlooked in viral genomes [11]. It is possible, in principle, to predict merely from sequence analyses whether a protein is expressed from an overlapping ORF, provided that the protein confers a beneficial function to the virus [12]. In that case, the additional selection pressure caused on the sequence of the reading frame that it overlaps results in a lower rate of synonymous codon substitution in that second frame [13,14].

Surveys of the B19V and PARV4 genomes reported such a lower rate in the region of VP1 corresponding to the X ORF [15], as well as in the region corresponding to ARF1 and ARF2 [10], but did not provide an estimate of the statistical significance of this reduction. In contrast, the dedicated software Synplot2 [16] can quantify the probability that an ORF with a reduced synonymous codon substitution rate is expressed and functional. Synplot2 has been extensively validated on experimentally proven overlaps and since its release was used to detect over 15 overlapping ORFs that were then confirmed experimentally (e.g., [17,18,19]). Interestingly, Synplot2 can also detect non-coding functional elements [16].

We thus chose to use Synplot2 to analyze the VP1 coding sequences of B19V and PARV4. Synplot2 detected several regions that may correspond either to protein-coding ORFs (including that of the X protein and of ARF1) or to non-coding functional elements. We compared the sequence properties of the erythroparvovirus X protein with that of tetraparvovirus ARF1 and determined that they were very similar.

## 2. Materials and Methods

### 2.1. Sequence Collection

We collected in Genbank (30 July 2019) the full coding sequences of VP1 for all isolates of viral species investigated here, restricting the search to viral sequences to avoid synthetic sequences. In particular, the sequence of reference erythro- and tetraparvoviruses are in Table 1. The recommendations for Synplot2 [16] specify to remove sequences containing insertions or deletions longer than 50 nucleotides with respect to the reference sequence, and those with <75% nucleotide similarity over 90% of the length of the query. No erythroparvovirus or tetraparvovirus sequence needed to be removed to comply with these recommendations, owing to their low sequence divergence.

### 2.2. Nucleotide Sequence Alignment and Analysis

To generate codon-respecting alignments based on the coding sequence of VP1, we used the program TranslatorX [20] with the “Muscle” option. The resulting codon-based alignments are in the Alignments S1–S4. Alignment S5 contains the potential start codons of the X ORF in all erythro- and tetraparvovirus species.

We looked for potential Internal Ribosome Entry sites (IRES) using IRESpy [21] and for ribosomal frameshifting sites using PRfect [22].

We used RNAz [23,24] to predict functional RNA structures on the basis of thermodynamic stability and evolutionary sequence conservation.

### 2.3. Detection of Regions with Lower Synonymous Substitution Rate

We used Synplot2 (version 2014-12-05) [16] to identify overlapping functional elements, with two sizes of sliding window: 25 and 45 codons. A window of 25 codons provides better specificity, which helped us identify how many regions have a decreased synonymous substitution rate; whereas a window of 45 codons provides better sensitivity, which helped us map the precise boundaries of the regions identified. We present Synplot2 plots computed with a window of either 25 or 45 codons, depending on which window size better shows the regions identified. The boundaries of these regions were always mapped with a window of 45 codons.

### 2.4. Protein Sequence Alignment and Domain Identification

All protein sequence alignments are presented using Jalview (version 2.11) [25] with the ClustalX coloring scheme [26]. We carried out phylogenetic analyses using phylogeny.fr [27] with default options (removal of poorly aligned regions with Gblocks [28] and phylogeny inference using PhyML (version 3.1) [29]). To add unaligned sequences into a reference alignment, we used MAFFT (version 7) with the --add option [30]. The S6 Alignment contains the sequence alignment of the X proteins and ARF1 proteins in text format (one representative for each *Erythroparvovirus* and *Tetraparvovirus* species). We used HHpred [31] ran in July 2023 against the databases PFAM (version 35) and PDB (version PDB_mmCIF30_18 Jun from the MPI toolkit [32]) to identify protein domains such as PLA2.

### 2.5. Prediction of Protein Structural Features

We used MetaDisorder [33] to predict disordered regions, in accordance with the principles described in [34], and DeepCoil [35] to predict coiled-coil regions.

We used two complementary methods to reliably predict transmembrane segments, as explained in [36]. First, we compared the predictions of several transmembrane prediction programs on a single protein, for each protein (“vertical” approach), by using CCTOP [37]. Second, we compared the prediction of a single program (TM-Coffee [38]) on several homologs (“horizontal” approach).

## 3. Results

### 3.1. The VP1 Gene of B19V Contains 3 Regions with Significantly Increased Synonymous Conservation, among Which the X ORF

To determine whether the VP1 gene of B19V might encode other proteins in overlapping reading frames, we collected the VP1 coding sequences (CDS) of all genotypes of B19V available in GenBank, translated them, aligned their amino acid sequences, and back-translated them to yield a nucleotide sequence alignment (see Section 2). Next, we determined whether the alignment contained regions with a reduced variability of substitutions at synonymous sites, using Synplot2 [16].

Synplot2 identified three regions with a statistically significant decrease in the variability of synonymous substitutions (Figure 2B). These regions are visible as peaks in Figure 2B, since Synplot2 actually plots the *increase* in *conservation* of synonymous substitutions instead of their *decrease* in *variability*.

The first region spans codons 58–163 of VP1 (see Table 2), and corresponds to the hypothetical X ORF (see Section 1). This ORF is devoid of stop codons in frame +1 relative to VP1 (Figure 2C) in all B19V sequences. A potential AUG start codon overlaps codon 84 of VP1 and is conserved in all B19V sequences, confirming that the X ORF has the potential to code for a protein. As Figure 2A shows, the X ORF is entirely embedded within the region encoding VP1u (the N-terminus of the capsid protein, found in VP1 but not in VP2), and partially overlaps the region encoding the Phospholipase A2 (PLA2) domain of VP1 [39,40].

Note that the reduction in synonymous variability starts quite upstream of the putative AUG start codon of the X ORF (at codon 58 and 84 of the VP1 gene, respectively, see Figure 2B and Table 2). This might indicate the presence of a regulatory region that enhances the translation of the X ORF, between codons 58 and 84, underlined in Figure 2B (see Section 4).

The X ORF is found in all other erythroparvovirus sequences in Genbank. (Table 1 lists the accession numbers of only the reference genome sequence of each species, and Alignment S6 presents the corresponding sequence of the X protein); see below for the special case of bovine parvovirus 3. An X ORF is also found in two erythroparvovirus-like sequences: one thought to be an endogenous virus [41], Indri endogenous parvovirus, and one of unknown status, Hyaena parvovirus-like sequence (Alignment S6).

The second region detected by Synplot2 spans codons 185–239 of VP1 (Figure 2B and Table 2), and has not been described yet, to our knowledge. We called it the “Y region”. Other erythroparvoviruses do not contain an equivalent region. The Y region overlaps the region of VP1 located downstream of the PLA2 domain and extends slightly into VP2 (Figure 2A). It is devoid of stop codons in frame +2 relative to VP1 in all B19V sequences (Figure 2C) and thus constitutes a potential CDS of 46 aa’s. However, it lacks a potential AUG start codon. It might thus either correspond to a non-coding functional element, or code for a protein expressed through a non-canonical mechanism.

Functional RNA elements often have a conserved secondary structure, but we could not detect the signature of such a structure in the Y region (using RNAz [23,24]). We could not either identify by sequence analysis potential sites that would direct non-canonical translation of the Y region, such as an IRES (Internal Ribosome Entry Site) [21] or a ribosomal frameshift [22], though such sites remain difficult to predict. Another possibility for expressing the Y region would be polymerase slippage [42], for which there is no consensus site. Note that all these non-canonical mechanisms would result in a chimeric protein composed of the N-terminus of VP1 fused to the Y coding sequence.

The third region detected by Synplot2 is located at the very C-terminus of the VP1 CDS (codons 771–781) (Figure 2B). It corresponds to the N-terminus of the 11 kDa protein (Figure 2A), known to be expressed in the +1 frame relative to VP1 from an AUG that overlaps codon 756 of VP1 [7,8].

Altogether, these data show that the VP1 gene of B19V encodes a protein, X, in the +1 frame. An X ORF is also present in all erythroparvovirus sequences. The VP1 gene of B19V also contains another region, Y, which corresponds either to a non-coding functional element or to the coding sequence of a protein in the +2 frame, which might be expressed as a fusion with the N-terminus of VP1.

**Figure 2 viruses-16-00191-f002:**
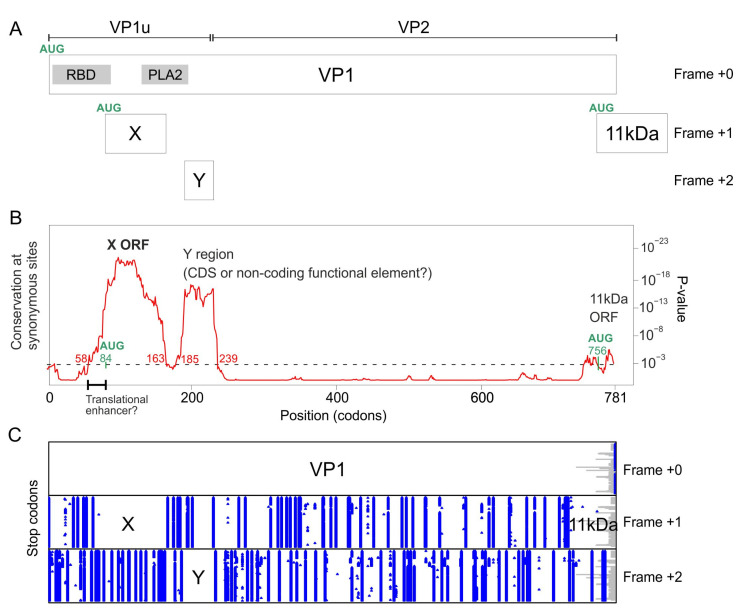
Synplot2 detects 3 regions with significantly lower synonymous variability in the VP1 coding sequence of B19V, among which the X ORF. (**A**) The VP1 gene and of its overlapping elements (coding sequences or functional elements). PLA2: Phospholipase A2 domain. RBD: receptor-binding domain [43]. VP1u: Vp1-unique region. (**B**) Sequence conservation at synonymous sites in an alignment of coding sequences of B19V VP1 (121 sequences ranging from 87% to 99% nucleotide identity), using a 25-codon sliding window. The plot corresponds to the *p*-value calculated by Synplot2 based on the number of substitutions observed versus the number expected under a null model (in which synonymous sites evolve neutrally). Regions in which synonymous substitutions are significantly decreased are indicated. The horizontal dotted line shows the significance cut-off value (10^−3^). Note that the first region with reduced synonymous variability starts markedly before the potential AUG start codon of the X protein (in green). This region, indicated by a thick line, might correspond to a functional element, which might facilitate the non-canonical translation of the X protein or the splicing of an X-specific RNA transcript (see text and Section 4). (**C**) Position of stop codons (blue) in the 3 potential frames, and gaps in alignment (gray) in the 121 sequences.

### 3.2. The VP1 Gene of PARV4 Contains 2 Regions with Significantly Reduced Synonymous Variability, Corresponding to ARF1 and ARF2

To determine whether the VP1 gene of PARV4 encodes other proteins in overlapping reading frames, we analyzed the VP1 coding sequence of all strains of PARV4 by using Synplot2, as described above for B19V. Figure 3B shows that two regions have a highly significant increase in the conservation of synonymous sites (Table 2):

The first region spans codons 180–263 of VP1 (Table 2), which corresponds to the ARF1 ORF [10] (see Section 1). ARF1 is devoid of stop codons in frame +1 relative to VP1 (Figure 3B) in all PARV4 sequences. It has a potential AUG start codon conserved in all PARV4 sequences, overlapping codon 187 of VP1. ARF1 is embedded within the VP1u region, and partially overlaps the PLA2 domain (Figure 3A). An ORF similar to ARF1 is found in all other tetraparvovirus sequences in Genbank (Alignment S6 presents the sequence of the ARF1 protein of one representative per species), with the exception of porcine parvovirus 2 (see below).

The second region detected by Synplot2 spans codons 294–397, and corresponds to the ARF2 ORF [10] (see Section 1). ARF2 is devoid of stop codons in frame +1 relative to VP1 (Figure 3C). It has a potential AUG start codon conserved in all PARV4 sequences, overlapping codon 294 of VP1. ARF2 overlaps the region of VP1 immediately downstream the PLA2 domain, and extends slightly into VP2 (Figure 3A). Note that PARV4 ARF2 and the putative Y protein of B19V are not homologous, since they are encoded in different frames relative to VP1 (respectively +1 and +2, compare Figure 2A and Figure 3A).

An ORF similar to ARF2 is found only in tetraparvoviruses closely related to PARV4: hokoviruses (porcine, bovine and ovine), and deer tetraparvovirus. We present their aa sequence in Appendix A. ARF2 has a predicted transmembrane segment near its N-terminus.

Altogether, these data show that the VP1 gene of PARV4 encodes a protein, ARF1, in the +1 frame. An ARF1 ORF is also present in all tetraparvovirus sequences. The VP1 gene also encodes a second protein, ARF2, in the +1 frame, conserved in tetraparvoviruses closely related to PARV4.

### 3.3. The B19V X Protein and PARV4 ARF1 Protein Have Similar Sequence Features

Next, we examined the predicted sequence features of B19V X protein and of PARV4 ARF1. Figure 4 presents multiple sequence alignments of the erythroparvovirus X protein (Figure 4A) and of tetraparvovirus ARF1 (Figure 4B). The erythroparvovirus X protein contains a predicted central transmembrane segment (Figure 4A), followed by a positively charged region, predicted to be inside the cytosol (“positive-inside rule” [44]). In B19V and the three closely related primate erythroparvoviruses, the C-terminus of the X protein is predicted to form a second transmembrane segment (boxed in Figure 4A). It is unusual that closely related proteins would vary in the number of transmembrane proteins they contain, and therefore these predictions should be taken with caution.

Tetraparvovirus ARF1 has a size and predicted organization similar to that of the X protein (compare Figure 4A,B), composed of an extra-cytosolic N-terminus, a central transmembrane segment, and a positively charged, intra-cytosolic region.

To further analyze the similarity between the X protein and ARF1, we examined in detail how their sequences align when based on the much more reliable alignment of VP1, and in particular its PLA2 domain. Indeed, PLA2 contains numerous strictly conserved amino acids (aa) [39,40], which makes its sequence alignment highly reliable.

To generate a reliable alignment of erythroparvovirus X proteins and tetraparvovirus ARF1 based on the alignment of VP1, we:(a)converted the aa alignment of the VP1 proteins into an alignment of nucleotide sequences by using TranslatorX [20];(b)translated this alignment in the reading frame of X and ARF1, i.e., +1 relative to VP1. This procedure is described graphically in a previous article [45].

The resulting alignment of X and ARF1 is shown in Figure 5A, while the reference alignment of VP1 is shown below, in Figure 5B. (Only the PLA2 domain of VP1 is shown, because the region upstream is not well conserved). As Figure 5A shows, the central transmembrane segments of X and ARF1 align together perfectly. Three aa positions are identical between X and ARF1, and one position is very similar (aromatic: Y, W or F). They are indicated above the alignment in Figure 5A.

Taken together, these data indicate that the central region of X and ARF1 is highly similar, consisting of a predicted transmembrane segment with several identical amino acids.

### 3.4. Conserved Features of the X and ARF1 Proteins Mostly but Not Exclusively Correspond to Conserved Motifs of the PLA2 Domain of VP1

Since the X and ARF1 ORFs partially overlap the PLA2 domain of VP1 (Figure 3A,B), we asked whether conserved sequence features of the X and ARF1 proteins are imposed by conserved sequence motifs of PLA2. As Figure 5B shows, the region of PLA2 overlapped by the X protein contains two conserved features: (1) the putative calcium (Ca^2+^)-binding loop (aa 130–134 in B19V); and (2) a region involved in the catalytic network, containing 3 strictly conserved aa’s (H153, D154 and Y157 in B19V) [39,40].

The conserved features of the X protein do correspond to these conserved features of PLA2. First, the transmembrane segment of the X protein overlaps the Ca^2+^-binding loop. Second, strictly conserved positions of the X protein (P43, L50, G73 in B19V, boxed in Figure 5A) overlap strictly conserved positions of PLA2, boxed in Figure 5B: P126 and P133 (both within the Ca^2+^-binding loop), and R156, close to conserved aa’s of the catalytic network. Likewise, the semi-conserved position of the X protein (Y54 in B19V) corresponds to a strictly conserved position of VP1 (L137 in B19V).

Clearly the PLA2 enzyme is under stringent selection pressure to conserve aa’s responsible for its catalytic activity. Therefore, one might assume that the sequence conservation within the X protein is dictated by PLA2. However, the sequence of strictly conserved aa’s of X is not completely imposed by PLA2. For example, consider the strictly conserved P133 and G134 in PLA2, which overlap the aa L50 strictly conserved in the X frame (Figure 5). Conservation of this Leucine is not imposed by the conservation of P133 and G134, since the dipeptide PG (Proline–Glycine) can be encoded by the nucleotides CCNGGN, in which N is any nucleotide. The first corresponding codon in the +1 frame relative to PLA2, CNG, can encode not only Leucine (CTG), but also Proline (CCG), Glutamine (CAG), or Arginine (CGG). Likewise, none of the conserved positions of the X protein are completely imposed by conservation of PLA2.

Altogether, these data show that conserved features of the X and ARF1 proteins mostly, but not exclusively, correspond to conserved motifs of the PLA2 domain of VP1.

### 3.5. The VP1 Gene of Bovine Parvovirus 3 and Porcine Parvovirus 2 Differs from That of Other Erythro- and Tetraparvoviruses

#### 3.5.1. Bovine Parvovirus 3 VP1 Gene Encodes an X-like ORF, despite Not Encoding a PLA2 Domain

We noticed that in one species, *Ungulate erythroparvovirus 1*, VP1 completely lacks the sequence signature of the PLA2 domain found in all other eythro- and tetraparvoviruses. This species is also commonly called bovine parvovirus 3 (bPARV3) [46], and is basal to the *Erythroparvovirus* phylogeny [46].

Synplot2 detected in the VP1 CDS of bPARV3 a region with significantly reduced synonymous variability, slightly upstream of the VP1/VP2 boundary (Figure 6B). This region corresponds almost exactly to an ORF conserved in all strains of bPARV3, in frame +1 relative to VP1 (Figure 6C). We called it “X-like” ORF, since it has a similar location as the X ORF of eythro- and tetraparvoviruses and has the potential to code for a protein with a similar length (99 aa’s) and organization (central transmembrane segment) as the X protein. The sequence of the bPARV3 X-like protein is shown in Figure 4C.

These data show that the bPARV3 VP1 gene encodes an X-like protein in the +1 frame despite not encoding a PLA2 domain (see also Section 4). Note that this cannot be due to a sequencing error in the bPARV3 VP1 gene, since there are 13 sequences for it (ranging from 92% to 99% identity) generated by several research groups.

#### 3.5.2. Porcine Parvovirus 2 Does Not Encode an X ORF, but Encodes a “Z ORF” Overlapping VP1

As mentioned above, there is no X-like ORF in porcine parvovirus 2 (pPARV2) (also called cnvirus [47]), which belongs to the species *Ungulate tetraparvovirus 3*, and is basal to the tetraparvovirus phylogeny [47]. We examined its VP1 coding sequence with Synplot2. Three regions have a significant increase in the conservation of synonymous sites (Figure 7B):

The first region spans codons 1–57. It is interrupted by stop codons both in +1 and +2 frames relative to VP1 (Figure 7C) and may thus correspond to a non-coding functional element.

The second region spans codons 193–309. It is devoid of stop codons in frame +1 relative to VP1 (Figure 7C) in all sequences of pPARV2, except one (accession number MK378188). It contains a potential AUG start codon overlapping codon 193 of VP1, conserved in all sequences. Thus, this region probably encodes a protein, which we called “Z protein”. The Z ORF overlaps the region of VP1 upstream the PLA2 domain and slightly extends into the N-terminus of PLA2 (Figure 7A). The sequence of the Z protein is shown in Appendix A. It has a rather low sequence complexity, and its N-and C-termini are predicted to be structurally disordered.

The third region spans codons 355–449. It is interrupted by stop codons both in frames +1 and +2 relative to VP1 (Figure 7C) and may thus correspond to a non-coding functional element.

Altogether, these data show that the pPARV2 VP1 gene encodes a protein (the Z protein), unrelated to the erythroparvovirus X protein or to tetraparvovirus ARF1, in the +1 frame. It also encodes two probable non-coding functional elements.

## 4. Discussion

### 4.1. Sequence Analyses Provide Compelling Evidence That the X Protein Must Be Expressed and Have a Crucial Function

The X ORF was noticed in B19V as early as 1986 [48], but has lived up to its name, since no experimental support has ever been provided for its translation or essentiality in infected cells. Indeed, substituting its presumed start codon by a stop codon had no effect on replication, infectivity, or capsid production in cells permissive for B19 [9].

An earlier sequence analysis provided hints that the product of the X ORF was functional, by detecting a decrease in synonymous codon variability in the region of overlap with VP1 [15], but could not determine whether this decrease was significant. Here we show that it is highly significant, using a dedicated software extensively validated on experimentally proven overlaps, Synplot2. In addition, we show that the X ORF is conserved in all erythroparvoviruses. Given the high rate of evolution of viruses, this provides compelling evidence that the X ORF must be expressed and play an essential function in the viral life cycle.

### 4.2. The X Protein Could Be Translated Either by a Non-Conventional Mechanism or from an Overlooked mRNA

The X ORF of B19V has a potential AUG start codon in all erythroparvoviruses, but cannot be encoded in a monocistronic fashion by any known viral mRNA (see Appendix A, panel A). These observations suggest that the X protein of B19V is translated either (1) by a non-canonical mechanism, or (2) from a currently unmapped mRNA. We briefly discuss both hypotheses, which we present only as a starting point to guide experimental approaches.

#### 4.2.1. Translation of the X ORF through a Non-Canonical Mechanism

In vertebrates, two main factors influence canonical translation: (1) the strength of the “Kozak sequence” surrounding the initiator AUG codon [49]; and (2) the position of the AUG codon in the mRNA. In general, translation initiates at the first AUG with an optimal Kozak sequence, but many exceptions are known (for a review, see the work presented in [50]). For example, a downstream AUG can sometimes initiate translation even if it is separated from the first optimal AUG by intervening AUGs, thanks to a mechanism called “re-initiation” (for a review, [51]). For example, in B19V, the VP1 AUG codon is preceded by 7 upstream AUG codons that form mini-ORFs (Figure 8), and is accessed by re-initiation after having first initiated translation at some of these mini-ORFs [52]. Note that the presence of these seven upstream AUGs severely decreases the translation level of VP1 [52].

In principle, the B19V X ORF might also be translated from the VP1 mRNA by re-initiation, since it is separated from the VP1 AUG start codon by 4 AUGs (Figure 8). However, the efficiency of translation would presumably be very low [51]. Interestingly, in B19V, the 77 nucleotides upstream of the presumed AUG start codon of the X ORF have a significantly reduced variability in synonymous codons (nt 172–250 of the VP1 CDS, see Figure 2B and Table 2, corresponding to nucleotides 2795–2873 of the genome, see Figure 8, bottom right). It is tempting to speculate that this region corresponds to a translation enhancer, i.e., a regulatory element that would enhance the translation efficiency of the X ORF.

#### 4.2.2. Translation of the X ORF from a Currently Unmapped mRNA

A second mechanism might in principle ensure translation of the X ORF: the existence of an unmapped mRNA, generated by an overlooked splice acceptor site. Two conditions would be required for a splice acceptor site to generate a monocistronic transcript that encodes the X protein of B19V: (1) this site should be conserved in all isolates of B19V; (2) it should be located in the region between the VP1 start codon and the presumed start codon of the X protein (nt 251–253 of the VP1 CDS).

We found 3 such potential sites (having the canonical sequence (C/U)AG preceded by a region rich in pyrimidines (C/U) [53]), at nucleotides 158–160, 185–187, and 231–233 of the VP1 CDS. (The respective coordinates of the acceptor G in the genomic sequence of B19V are 2783, 2810 and 2856, see Figure 8). Each acceptor site would yield a monocistronic transcript that encodes the X ORF, by splicing out both the VP1 AUG start codon and the 4 following AUG codons located upstream of the presumed AUG start codon of the X protein (in red in Figure 8). Interestingly, these potential splice acceptor sites are located in or near in the potential regulatory region immediately upstream of the X ORF (Figure 8, bottom right), which has a decreased synonymous variability (Figure 2B and Table 2). The resulting hypothetical transcript, which might encode the X ORF in a monocistronic fashion, is depicted in Appendix A, panel A.

### 4.3. Are X, ARF1, and the Dependoparvovirus MAAP Protein Likely to Have a Similar Function?

The X and ARF1 ORFs overlap the same location of the VP1 gene (corresponding to the PLA2 domain), in the same frame (+1) relative to VP1, and encode proteins with similar properties. In the genus *Dependoparvovirus*, another ORF, MAAP, whose translation has been proven experimentally, also overlaps the same region of the VP1 gene in the +1 reading frame [54,55]. This similarity begs the question: do X, ARF1, and MAAP have a common origin and a similar function? We cannot answer the first question (origin) here, and only provide partial clues for the second (function).

Indeed, the evolutionary history of the ORFs overlapping the region encoding PLA2 appears very complex. First, a phylogenetic tree based on the PLA2 domain of VP1 (Figure 9) indicates that erythroparvovirus sequences are somewhat related to dependoparvovirus sequences, while the placement of tetraparvoviruses is uncertain (low bootstrap value (0.22; a node with bootstrap value inferior to 0.7 is considered unreliable). Second, in the genus *Copiparvovirus*, preliminary analyses (not presented here) suggest that some, but not all species also encode an uninterrupted ORF similar to X and ARF1 (i.e., that overlaps in the +1 frame the region of the VP1 gene encoding the PLA2 domain). Therefore, determining whether the X, ARF1, MAAP, and copiparvovirus ORFs have a common origin and elucidating their evolutionary history will require further studies, which are clearly outside of the scope of this work.

However, we note that the predicted sequence features of MAAP differ radically from those of X and ARF1: the region of MAAP that overlaps PLA2 is disordered and T/S-rich, while in X it contains a transmembrane region [54]. Thus, regardless of whether they have a common origin, there is no rationale for thinking that the MAAP protein has a similar function to X or to ARF1.

### 4.4. Neither ARF1 nor the X Protein Are Homologous to the Protoparvovirus SAT Protein

An earlier work [10] hypothesized that PARV4 ARF1 was homologous to the SAT protein, another short, transmembrane protein encoded in the +1 frame of the VP1 gene in the genus *Protoparvovirus* [56]. However, SAT and ARF1 cannot have a common origin, since SAT is encoded by the N-terminus of VP2, downstream of the region encoding the PLA2 domain (our observations), unlike PARV4 ARF1 and the B19V X protein, which overlap PLA2 (see Figure 2 and Figure 3).

## 5. Conclusions

While a systematic effort has been made to discover overlapping genes in RNA viruses by sequence analyses [16], this has not yet been the case in DNA viruses. Our findings confirm (if that were needed) that overlapping genes remain to be discovered in DNA viruses (we know of at least another case already flagged by sequence analyses, in human bocavirus [57]). We encourage all virologists who sequence genomes to look for overlapping genes using the simple tools and strategies presented here, as well as other available software to predict overlapping genes, which are complementary (for a review, [12]). This is perfectly feasible if you are a bench virologist lacking programming skills (like the author), since all analyses were performed using web-based, relatively user-friendly programs (see Section 2) on a standard laptop computer.

## Figures and Tables

**Figure 1 viruses-16-00191-f001:**
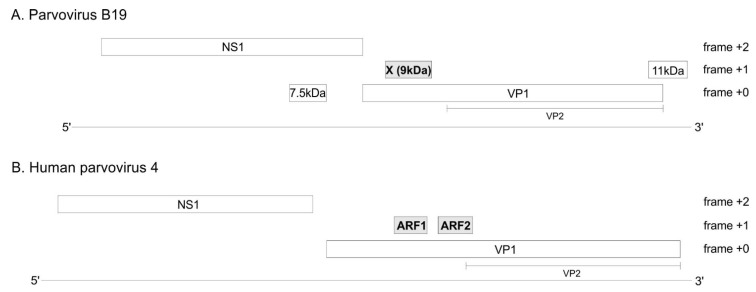
B19V (**A**) and PARV4 (**B**) encode three suspected protein-coding ORFs. Long, horizontal lines represent the viral genomes. Boxes represent ORFs (open reading frames). The three ORFs suspected to code for a protein are in grey. The VP2 isoform of VP1 is represented under VP1.

**Figure 3 viruses-16-00191-f003:**
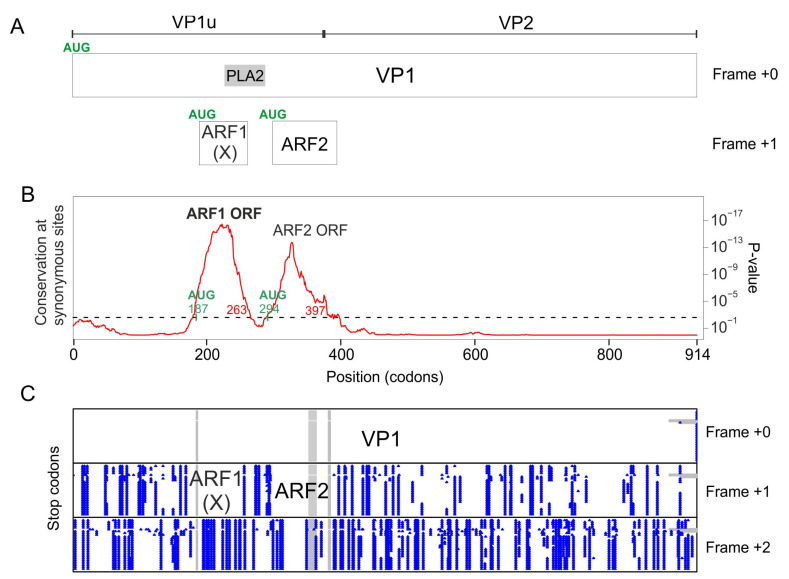
Synplot2 detects 2 regions with significantly lower synonymous-site variability in the VP1 coding sequence of PARV4. (**A**) Conventions are as in Figure 2. (**B**) Conservation at synonymous sites in an alignment of coding sequences of PARV4 VP1 (21 sequences ranging from 93% to 99% identity), using a 25-codon sliding window in Synplot2. (**C**) Position of stop codons (blue) in the 3 potential frames, and gaps in alignment (gray) in the 21 sequences.

**Figure 4 viruses-16-00191-f004:**
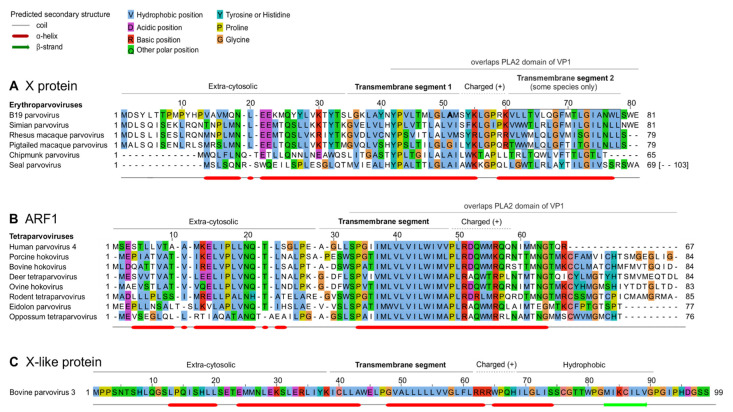
Similar organization of the erythroparvovirus X protein, tetraparvovirus ARF1, and bovine parvovirus 3 X-like protein. (**A**) Multiple sequence alignment of the erythroparvovirus X proteins. Numbering above the alignment corresponds to B19V. The sequences presented assume that the first AUG of each X ORF is used to initiate translation. PLA2: Phospholipase A2 domain. (**B**) Alignment of the tetraparvovirus ARF1 proteins. Numbering corresponds to PARV4. (**C**) Sequence of the X-like protein of bovine parvovirus 3.

**Figure 5 viruses-16-00191-f005:**
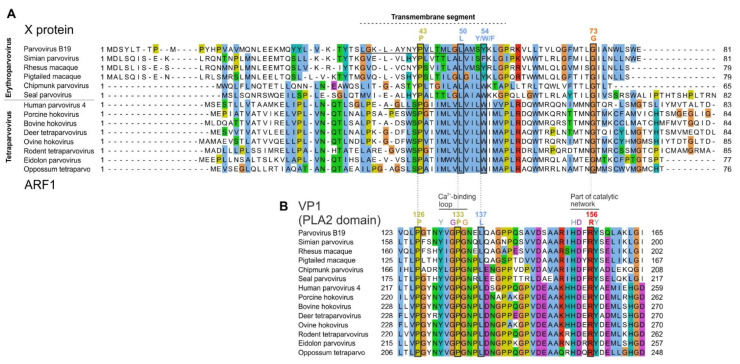
Alignment of all X and ARF1 proteins based on the reference alignment of the PLA2 domain of VP1. Conventions are as in Figure 4. (**A**) Alignment of the X protein of erythro- and tetraparvoviruses, derived from the reference alignment of VP1 presented in panel (**B**). The X alignment was generated from the VP1 alignment by using TranslatorX [20] (see text). Strictly or semi-conserved aa’s are indicated. Predicted transmembrane regions are underlined in the sequence of B19V X and PARV4 ARF1. (**B**) Reference alignment of VP1. Only the reliably aligned N-terminal part of the PLA2 domain is shown. Thin vertical lines show the correspondence between aa’s encoded by overlapping codons in the X frame (panel (**A**)) and in the VP1 frame (panel (**B**)). Aa’s that overlap conserved positions of the X protein are boxed. Other conserved aa’s involved in functional elements of PLA2 are also indicated.

**Figure 6 viruses-16-00191-f006:**
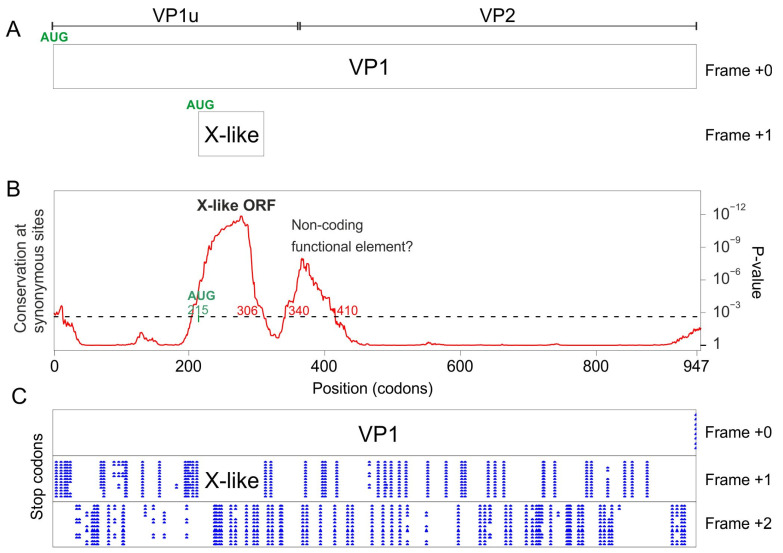
Synonymous-site variability in the VP1 coding sequence of bovine parvovirus 3 (bPARV3). (**A**) Conventions are as in Figure 2. The location of the VP1/VP2 boundary is only inferred. bARV3 VP1 contains no PLA2 domain, unlike all other erythro- and tetraparvoviruses (see text). (**B**) Conservation at synonymous sites in an alignment of the coding sequences of bPARV3 VP1 (13 sequences ranging from 92% to 99% identity), using a 45-codon sliding window in Synplot2. (**C**) Position of stop codons (blue) in the 3 potential frames, and gaps in alignment (gray) in the 13 sequences.

**Figure 7 viruses-16-00191-f007:**
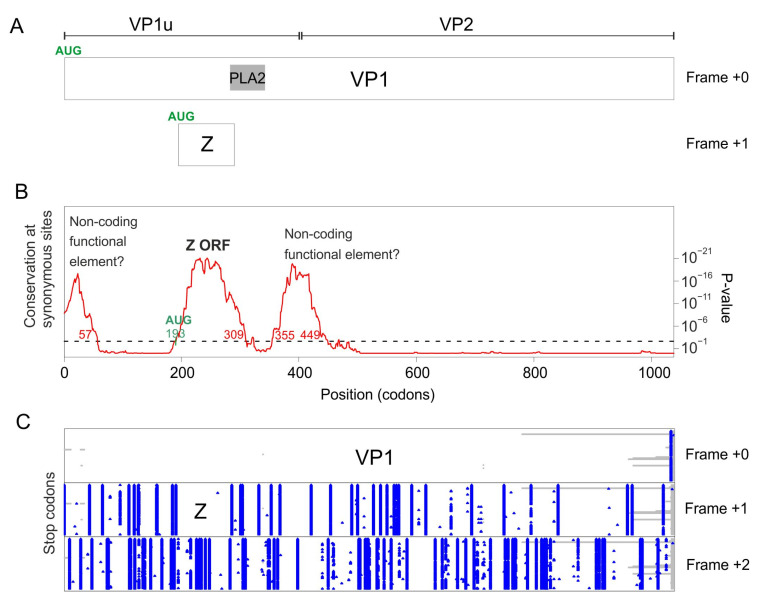
Synonymous-site variability in the VP1 coding sequence of porcine parvovirus 2. (**A**) Conventions are as in Figure 2. The location of the VP1/VP2 boundary is only inferred. (**B**) Conservation at synonymous sites in an alignment of the coding sequences of pPARV2 VP1 (90 sequences ranging from 93% to 99% identity), using a 45-codon sliding window in Synplot2. (**C**) Position of stop codons (blue) in the 3 potential frames, and gaps in alignment (gray) in the 90 sequences.

**Figure 8 viruses-16-00191-f008:**
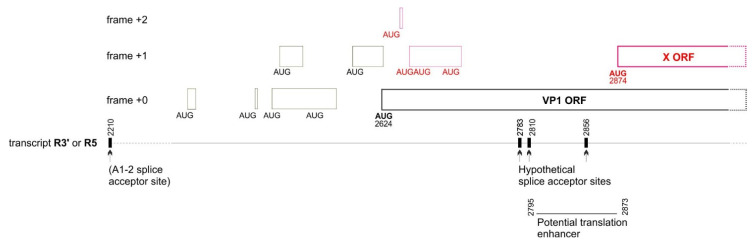
Elements that could influence translation of the VP1 and X ORFs of B19V: upstream mini-ORFs, potential splice acceptors, and potential translation enhancer. The nomenclature of transcripts and splice sites is as in the work presented in [4]. Thin boxes represent mini-ORFs. The mini-ORFs in black influence the translation of VP1 [51], and might also influence that of the X protein. The mini-ORFs in red are expected to influence the translation of the X protein but presumably not that of VP1. The region immediately upstream of the X ORF has a decreased synonymous variability (see Table 2 and Figure 2B), suggesting it has a regulatory function and might act as a translation enhancer.

**Figure 9 viruses-16-00191-f009:**
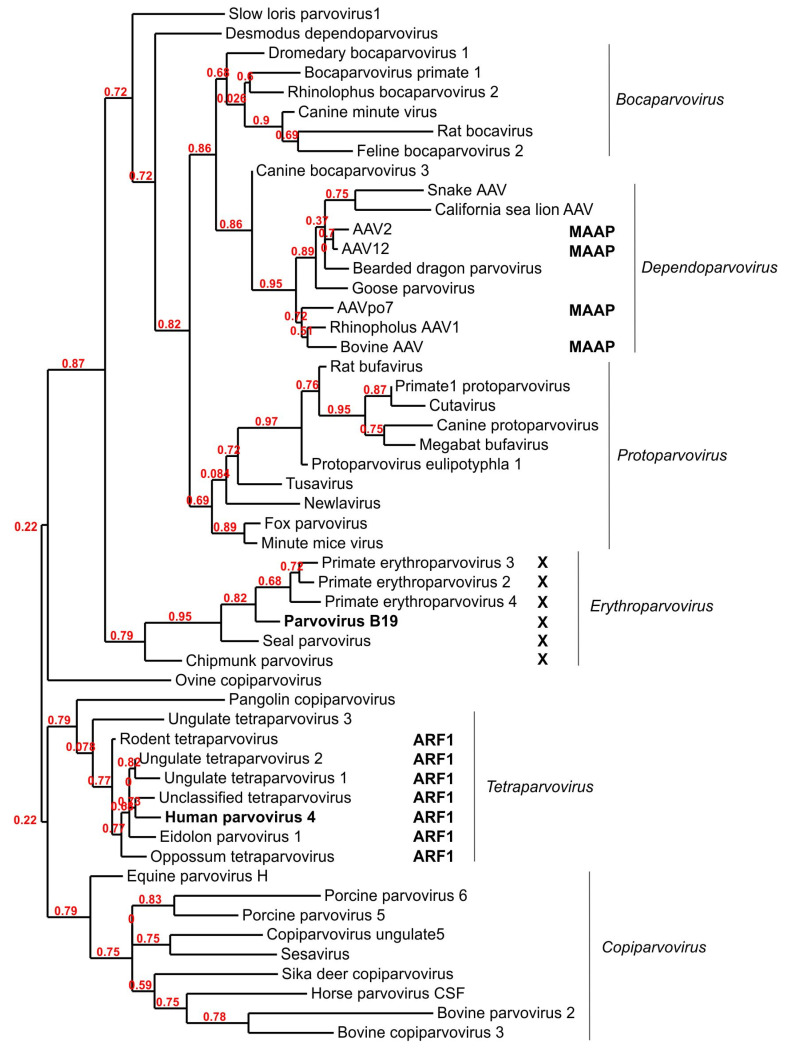
Phylogenetic analysis of *Erythro*- and *Tetraparvoviruses* and of related genera, based on the PLA2 domain. The sequences used for this phylogenetic tree are in the S7 alignment. Bootstrap values are indicated in red. Note that bPARV3 is not included in the analysis, since it has no PLA2 domain (see text). AAV: adeno-asssociated virus.

**Table 1 viruses-16-00191-t001:** Nucleotide sequences of virus species analyzed in this work.

Genus	Species	Common Name(s) [Abbreviation]	Genbank Genome Accession Number	Boundaries of the X ORF in the Genome Sequence (in Nucleotides)
*Erythroparvovirus*	*Primate erythroparvovirus 1*	**Parvovirus B19** **[B19V]**	NC_000883	2874–3119
*Erythroparvovirus*	*Primate erythroparvovirus 2*	Simian parvovirus	U26342.1	2718–2963
*Erythroparvovirus*	*Primate erythroparvovirus 3*	Rhesus macaque parvovirus	AF221122.1	2841–3080
*Erythroparvovirus*	*Primate erythroparvovirus 4*	Pig-tailed macaque parvovirus	AF221123.1	2563–2802
*Erythroparvovirus*	*Rodent erythroparvovirus 1*	Chipmunk parvovirus	GQ200736.1	3031–3228
*Erythroparvovirus*	*Pinniped erythroparvovirus 1*	Seal parvovirus	KF373759.1	2789–3100
*Erythroparvovirus* (*)	*Ungulate erythroparvovirus 1*	**Bovine parvovirus 3 [bPARV3]**	NC_037053	2627–2926
*Tetraparvovirus*	*Chiropteran tetraparvovirus 1*	Eidolon helvum parvovirus	NC_016744.1	2829–3062
*Tetraparvovirus*	*Primate tetraparvovirus 1*	**Human parvovirus 4** **[PARV4]**	NC_007018.1	2937–3140
*Tetraparvovirus*	*Ungulate tetraparvovirus 1*	Bovine hokovirus 1	NC_028136	2857–3111
*Tetraparvovirus*	*Ungulate tetraparvovirus 2*	Porcine hokovirus	EU200677.1	2808–3062
*Tetraparvovirus*	*Ungulate tetraparvovirus 5*	Deer tetraparvovirus	NC_031670.1	2766–3020
*Tetraparvovirus* (*)	*Ungulate tetraparvovirus 3*	**Porcine parvovirus 2 [pPARV2];** Porcine cnvirus; Parvovirus YX	NC_035180	No X ORF; boundaries of the Z ORF are 2817–3098
*Tetraparvovirus*	*Ungulate tetraparvovirus 4*	Ovine hokovirus	JF504699.1	2855–3112
*Tetraparvovirus*	-	Opossum parvovirus	MG745671.1	2862–3092
*Tetraparvovirus*	-	Rodent parvovirus	MG745669.1	2960–3217

The main species analyzed here are in bold. (*) The taxonomic classification of these species might need a revision in view of our analyses.

**Table 2 viruses-16-00191-t002:** Boundaries of VP1 regions with significantly lower synonymous codon variability (identified by Synplot2) encompassing potential protein-coding ORFs.

Virus Name	Region	Boundaries of the Region with Lower Synonymous Codon Variability in the VP1 CDS	Boundaries of the Corresponding ORF in the VP1 CDS
Parvovirus B19	X ORF	Codons 58–163 (nucleotides 172–489)	Codons 84–166 (Nucleotides 251–496)
Parvovirus B19	Y region ^(^*^)^	Codons 185–239 (nucleotides 553–715)	Codons 185–230 ^(^*^)^ (nucleotides 553–688)
Human parvovirus 4	ARF1 ORF (=X ORF)	Codons 180–263 (nucleotides 538–789)	Codons 187–255 (nucleotides 560–763)
Human parvovirus 4	ARF2 ORF	Codons 294–397 (nucleotides 880–1189)	Codons 295–379 (nucleotides 884–1135)
Bovine parvovirus 3	X-like ORF	Codons 205–306 (nucleotides 614–916)	Codons 215–315 (nucleotides 644–943)
Porcine parvovirus 2	Z ORF	Codons 193–309 (nucleotides 577–927)	Codons 193–285 (nucleotides 578–854)

(*): this region contains an ORF devoid of stop codon, but lacks a potential AUG start codon, and might not code for a protein.

## Data Availability

Data are contained within the article and Appendix A.

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
