# Peer review of "Parvovirus B19 and Human Parvovirus 4 Encode Similar Proteins in a Reading Frame Overlapping the VP1 Capsid Gene"

_viruses, 2024, doi:10.3390/v16020191_

Round 1
Reviewer 1 Report
Comments and Suggestions for Authors
In the current manuscript, the author evaluates the double coding of the VP1 region in parvovirus B19 and human parvovirus 4 using different bioinformatics approaches. The presented bioinformatic analysis is very detailed and comprehensive. The author gives strong evidence from multiple points of view, that both viruses have overlapping ORF in identical positions in respect to VP1 and that these ORFs most likely encode an expressed protein. Evolutionary conservation of these ORFs in the used set of Erythroparvoviruses and Tetraparvoviruses further supports this conclusion. Further analysis of ARF1 (Tetraparvoviruses) and protein X (Erythroparvoviruses) shows that they most likely have similar biochemical properties.
The Introduction is comprehensive enough to understand the background of the study.
From the presented analyses, it is clear that these are indeed regions that encode short proteins both in +1 reading frame when compared to VP1.
However, the evidence that these two proteins (ARF1 and X protein) are homologous (having a common ancestor) is not as clear.
Major questions.
The author should explain as to why at least two lines of analyses have not been performed (or not described in the article). First - phylogenetic trees, and second – protein 3D structure prediction.
For example, comparing properties of the phylogenetic tree of parvoviridae VP1 proteins (or Tetraparvovirus and Erythroparvovirus tree with some of its closest neighbours and more distant outgroups) with properties of X-protein/ARF1/X-like protein tree.
Minor comments:
The author should describe how many Tetraparvovirus and Erythroparvovirus genomes (species) were excluded due to limitations described in the Material and Methods section. It should be noted that readers should not draw the conclusion that ALL Tetraparvoviruses and Erythroparvoviruses encode these proteins due to the possibility of excluding early branching species. They might encode them, but this cannot be concluded from the data that has been presented yet. The author should also fix this in discussion (page 4, lines 135-137). The readers must be sufficiently warned. So, the legend for Figure 1 should clarify that only sequences/species used in this study are shown in the cladogram (and not all Tetraparvovirus or Erytroparvovirus genomes/species in the databank). Why not include all Tetraparvoviruses and Erythroparvoviruses in the analyses, at least for studying the presence of ORF in the respective position. As estimated by Campbell et al (https://doi.org/10.1371/journal.pbio.3001867) both Tetra- and Erythroparvoviruses are about 100 million years old. Conservation of ARF1 and protein X over all Tetra- and Erythroparvoviruses will further strengthen the author’s statements.
Two extra situations are also described, porcine parvovirus 2 does not have X ORF or ARF1 ORF and Bovine parvovirus 3 does not have PLA2 domain but still has X-like protein. Sometimes databases have errors in sequences, which are widely known in respective societies but can be still found in databases. Sometimes an in-depth evaluation of sequences helps to get out of annotation exceptions. This kind of analysis should be done (involving the experts of the respective viral taxon). If this type of control has been done then it should be noted in the manuscript.
As HHpred update regularly their protein domain HMMs, could you please specify when the HHpred was used to detect domains in respective sequences. And provide the version numbers of used bioinformatics tools if available.
In Conclusions, the author encourages virologists to use bioinformatics tools. In case of predicting double coding, the performance of different approaches strongly depends on the situation (age of viral taxon under study, strength of selective pressure etc). Depending on the situation the best algorithm may vary. So, please encourage to use all available double coding prediction algorithms because this is the best practice.
Reviewer 2 Report
Comments and Suggestions for Authors
This a neat and interesting paper that uses computational methods to examine the genomic and evolutionary characteristics of proteins encoded by short open reading frames (ORFs) overlapping the capsid proteins of two distinct parvovirus genera (erythroparvirus and tetraparvovirus). The author makes a convincing argument that two of these proteins - X (erythroparvovirus) and ARF (tetraparvovirus) are conserved under purifying selection and thus likely functional. Moreover, he presents evidence that X and ARF1 are homologous and have a common evolutionary origin. These observations may be useful in advancing understanding of parvovirus biology.
Major points
The paper makes no mention of the MAAP protein of dependoparvoviruses, which is also encoded by an overlapping frame in capsid. Given that the dependoparvovirus genus is fairly closely related to the erythro- and tetraparvovirus genera, and has been examined experimentally, the author should consider exploring any possible links between MAAP, X, and ARF1.
Minor points
Figure 5B - to my eye, the alignment looks as though it could be improved/simplified around positions 26-30. Interesting to note that in porcine hokovirus this region corresponds to a viral late domain (PSAP)
The page numbering has gone awry following the landscape oriented page 9
P10 Line three - "three lines of evidence" not "3 lines of evidence"
Round 2
Reviewer 1 Report
Comments and Suggestions for Authors
The author give answers to all reviewers questions. Either in Cover Letter or in modified manuscript. He also give explanations why further work (addressing next interesting questions) is out of the scope of current manuscript.
Several typos have to be fixed (like typos in abbreviations of Figure 6 legend and some more).